# A Suggested Vacuum Bagging Process for the Fabrication of Single-Walled Carbon Nanotube/Epoxy Composites That Maximize Electromagnetic Interference Shielding Effectiveness

**DOI:** 10.3390/polym13111867

**Published:** 2021-06-04

**Authors:** Min Ye Koo, Hon Chung Shin, Jonghwan Suhr, Gyo Woo Lee

**Affiliations:** 1Division of Mechanical Design Engineering, Jeonbuk National University, 567 Baekje-daero, Deokjin-gu, Jeonju 54896, Korea; mykoo@jbnu.ac.kr (M.Y.K.); chung5505@kcarbon.or.kr (H.C.S.); 2Division for Commercialization & Standardization, Korea Carbon Industry Promotion Agency, Unam-ro, Deokjin-gu, Jeonju 54853, Korea; 3School of Mechanical Engineering, Sungkyunkwan University, 2066 Seoburo, Jangan-gu, Suwon 16419, Korea; suhr@skku.edu; 4Department of Polymer Science and Engineering, Sungkyunkwan University, 2066 Seoburo, Jangan-gu, Suwon 16419, Korea

**Keywords:** single-walled carbon nanotubes (SWCNTs), SWCNT prepregs, vacuum-bagging process, electrical property, electromagnetic interference shielding effectiveness (EMI SE), skin depth theory

## Abstract

We designed and tested a manufacturing process that resulted in the formation of composites with maximized electrical conductivity and optimized electromagnetic interference (EMI) shielding effectiveness (SE) properties. Single-walled carbon nanotube (SWCNT) paper, which is a microscopic aggregate of van der Waals force interaction, was impregnated with semi-cured epoxy to make SWCNT prepregs. These prepregs were completely cured into SWCNT/epoxy composites. Fabricating and curing processes were executed under proper temperature cycle depending on the time. We inspected SWCNT paper and the interfacial state between the SWCNTs and epoxy in the composite with a field emission-scanning electron microscopy and calculated the SWCNT weight fraction through thermogravimetric analysis measurements. Using these observations, electrical conductivity and EMI SE were investigated according to thickness which could be controlled by the suggested manufacturing process as 1-, 5- and 10-layer composites. Finally, we determined ideal composite thickness and the associated number of prepreg layers using skin depth theory.

## 1. Introduction

The versatile properties and high aspect ratios of carbon nanotubes (CNTs) have attracted attention among researchers for their potential use as conductive fillers. Single-walled carbon nanotubes (SWCNTs), a type of CNTs that consist of single graphite sheets rolled into tubes, in particular, demonstrate impressive electrical conductivity at 10^6^–10^7^ S/m, a level which is capable of providing electrical properties to a polymer matrix [1,2,3]. Moreover, the SWCNTs prices have fallen as production technology has improved, bringing commercialization within reach. Many researchers have attempted to take advantage of the electrical conductive properties of SWCNTs to improve the shielding effectiveness (SE) of materials subject to electromagnetic interference (EMI) [4,5,6,7]. High electrical conductivity plays a key role in enhancing EMI SE. High loading of CNTs into the epoxy matrix is one of the most effective means of improving electrical conductivity. EMI SE can also be enhanced by precisely controlling a material’s thickness, with thicker materials more effectively blocking EMI. However, since thick objects are heavy and expensive to manufacture, a composite material that provides optimum EMI SE is required in consideration of thickness or mass. In sum, the materials that shield most effectively against EMI are those that have high electrical conductivity and are of optimal thickness.

Researchers are aware of problems with the interface between the CNTs and epoxy matrix, including weak interfacial bonds, high surface resistance, low compatibility, and a low level of CNT dispersion due to agglomeration by van der Waals forces in the epoxy matrix [8,9,10,11,12]. To overcome these problems, various alternatives have been proposed. CNTs are generally dispersed in the epoxy matrix through either chemical or physical methods [13,14,15,16]. Chemical methods require the introduction of functional groups through chemical treatment of or use of a surfactant on CNT surfaces. Physical methods include shear mixing and ultra-sonication to disperse bundled or randomly aggregated CNTs throughout an epoxy matrix. Chemical and physical dispersion methods each have their place, depending on the precise experiment being run. However, regardless of which method is used, there is a limit to the dispersion of CNTs, up to 5 wt% or more, in an epoxy matrix, as well as in a typical polymer matrix. The electrical properties of composites with low filler concentrations are unsatisfactory [9,12]. 

Reinforcing CNTs at high concentrations in an epoxy matrix is effective at transferring their properties to epoxy. However, once a large amount of CNTs are directly added to the epoxy matrix, the rapid increase in viscosity makes it difficult to fabricate a specimen. To obtain fine composites with a high CNT concentration, CNT paper may be impregnated using epoxy resin. CNT paper, also referred to as bucky paper, is often a thin sheet taken from a microscopic aggregate of CNTs that interacts with van der Waals forces. This paper is classified on the basis of its internal structure, with one variation of this paper consisting of randomly oriented CNTs with strong entanglement behavior [17,18,19,20,21,22], and the other consisting of aligned-orientation CNTs with a highly dense structure [23,24,25,26,27,28,29,30,31,32]. 

Impregnating the epoxy matrix into the CNT paper is difficult, as the epoxy resin must infiltrate the areas between individual CNTs. Wang et al. [21] suggested a fabrication process for the infiltration of acetone-diluted epoxy resin into SWCNT paper that relies on completely curing the composites using a hot press. Under these conditions, acetone-diluted epoxy resin has a low enough viscosity to infiltrate the SWCNT paper. Through this process, SWCNT-reinforced epoxy composites up to 39 wt% were achieved. Ashrafi et al. [22] compared two different epoxy impregnation methods—a vacuum-infiltration method and a hot-compression method to fabricate SWCNT-reinforced epoxy composites. They found that a specimen manufactured through vacuum-infiltration showed a higher Young’s modulus due to the lower void fraction than one generated by the hot-compression method. Ogasawara et al. [24] produced multi-walled CNT (MWCNT) paper using vertically aligned arrays and then impregnated the semi-cured epoxy resin film through a hot-melt prepreg method, ultimately obtaining approximately 28 wt% MWCNT-reinforced epoxy composites.

We attempted to optimize the manufacturing process for a high loading of SWCNTs reinforced in epoxy composites using a vacuum bagging system. The resulting products demonstrated high electrical conductivity and were relatively thin composites, both factors allowing the EMI SE of the material to be maximized. We measured EMI SE using different numbers of stacking layers to calculate the ideal thickness according to the skin depth theory. We were able to produce a composite that exhibited high EMI SE by a vacuum bagging system that was designed to execute a proper temperature cycle depending on the time. Furthermore, we applied the skin depth theory to determine the ideal number of prepreg layers.

## 2. Materials and Methods

### 2.1. Materials

The SWCNTs (Tuball, Ocsial Inc., Novosibirsk, Russia) were used as received without any further purification, and they were approximately 1.8 nm in average diameter, 5 μm in length, 75 wt% in purity, and 1.8 g/cm^3^ in density. An ionic liquid, 1-butyl-3-methylimidazolium tetrafluoroborate (BMIBF_4_, Sigma-Aldrich Inc., St. Louis, MO, USA) was used as a dispersant. Eighteen grams per square meter epoxy (TERA Engineering Inc., Yongin, Korea) was uniformly coated in a semicuring state on the releasing film.

### 2.2. Fabrication Methods

First, a sheet of SWCNT paper was prepared. An agate mortar was used to grind 0.4 g of SWCNTs, 3 mL of BMIBF4, and 10 mL of ethanol for 15 min. The mixed slurry was dispersed in 500 mL of ethanol using a horn-type sonicator (280 W, Ulsso Hitech Inc., Cheongju, Korea) for 48 min, which included six repetitions of 5 min executions and 3 min rests. The dispersed SWCNT colloid was filtered with cellulose filter paper (0.2 µm pore size, Advantec MFS Inc., Dublin, CA, USA). The remaining BMIBF_4_ was completely rinsed with ethanol. Undesired ethanol was then removed in an oven at 120 °C for 12 h. The SWCNT paper consisted of only SWCNTs with a diameter of 175 mm and a thickness of 50 μm. 

The vacuum bagging process was then used to infiltrate the epoxy in its semi-cured state into the SWCNT paper. The vacuum bagging method is typically defined as a system generating pressure and controlling the temperature of materials for fabricating composites. System equipment removes the trapped air or makes the internal structure denser when the composites are fabricated. Vacuum bagging systems are designed to use a breather and release film (Figure 1). An absolute pressure of 30.4 kPa was maintained during the cycle. The temperature cycle was controlled depending on the time as shown in Figure 2. Each cycle consisted of three steps. The SWCNT prepreg was made through steps 1 and 2. The prepreg was an intermediate material made of SWCNT paper and semi-cured epoxy. The epoxy matrix in this step had a low viscosity as a result of the increased temperature, allowing it in its semi-cured state to efficiently infiltrate the space between the SWCNTs in the SWCNT paper. The cycle we designed was based on a proper thermal rate and isothermal temperature. After finishing step 2 and starting step 3, we stacked SWCNT prepreg in layers of one, five, and ten. Then, we perfectly cured it after finishing step 3. Finally, we could obtain the SWCNT-reinforced epoxy composite film. To obtain a better perspective on the strengths of our process, SWCNT paper not infiltrated with epoxy was also produced and compared to composite samples.

We elected to manufacture SWCNT prepreg using the vacuum bagging method for several reasons. We expected that low-viscosity epoxy would be uniformly infiltrated in the SWCNT paper, which would lead to a high-loaded SWCNT composite, improving electrical conductivity and EMI SE. Furthermore, trapped air that is not removed at this stage remains in the voids, impairing performance when the prepregs are fabricated into composites. In particular, the prepreg can be stacked as much as desired, which also facilitates the control of composite thickness. 

EMI SE is improved with increases in electrical conductivity and composite thickness. Through our manufacturing process, we improved the electrical conductivity of high-loaded SWCNT composite and achieved improved control over thickness.

### 2.3. Measurements

SWCNTs were observed using a transmission electron microscope (TEM, JEM-2200FS, Tokyo, Japan). A field emission scanning electron microscope (FE-SEM, SU8220, Hitachi Inc., Tokyo, Japan) was used to analyze the impregnation of the epoxy into the SWCNTs. Thermogravimetric analysis (TGA) was performed to measure the contents of the SWCNTs and epoxy. The processing temperature was increased to 800 °C from room temperature at a rate of 10 °C/min and an Ar flow rate of 50 sccm. Electrical conductivity and sheet resistivity were measured with a 4-point probe method (RSD-1G, Dasol Eng Inc., Cheongju, Korea). EMI SE (E5061B, Keysight Technologies Inc., Santa Rosa, CA, USA) was investigated in accordance with ASTM D4935. The averages and deviations were calculated by measuring five times per sample of three specimens. In the case of specific EMI SE, those were calculated using the averages of EMI SE.

## 3. Results and Discussion

### 3.1. Morphology

The SWCNTs used in this study were single graphite sheets rolled into tubes. SWCNTs are approximately 1.8 nm in diameter and mostly exist as bundles as a result of van der Waals interactivity between the tubes. We performed TEM and FE-SEM to characterize the morphology of the SWCNT paper and SWCNT-reinforced epoxy composite. First, Figure 3a shows a TEM image of the SWCNTs. Bundles of SWCNTs were observed, and the diameters of each tube were 2 nm or less. On the basis of the TEM image, the tube might be seen as single-walled, and it fits well with the diameter suggested by the manufacturer. Figure 3b illustrates that SWCNT paper was successfully fabricated by the creation of self-assembled SWCNT networks during the vacuum filtration process. However, randomly-entangled SWCNT bundles can cause undesirable air fractions, and impair the electrical properties of the SWCNT paper or composite [30,31]. Developing a method of controlling or eliminating these unnecessary spaces is critical to improving the properties of SWCNT composites. The surfaces of our produced SWCNT-reinforced composite are shown in Figure 3c, an examination of which reveals that epoxy film properly infiltrated the SWCNT paper without epoxy aggregation. Furthermore, SWCNT bundles were wrapped in the epoxy during the vacuum bagging process. The original SWCNT bundles in the paper were approximately 17.4 nm in diameter, while epoxy-infiltrated SWCNTs were approximately 47.8 nm, an increase of 3 times. The epoxy was also wrapped around the SWCNTs as a result of the temperature cycle optimized for time, and most of the voids in Figure 3b were filled with epoxy.

Kim et al. [31] studied the mechanical properties of an MWCNT forest and found that it had an air fraction of approximately 98.1%. The SWCNT paper used in this study differs from the MWCNT forest, but since the CNTs used in both studies were quite dense we assume that the air fractions are similar. Our process filled the air space with a minimal amount of polymer, allowing for the natural fabrication of high-content, highly-conductive, shape-conserved SWCNT-reinforced epoxy composites.

### 3.2. TGA

The highly loaded SWCNT-reinforced epoxy composites were analyzed by TGA to calculate the precise amount of filler. SWCNT paper, epoxy film, and their composites were also tested, with results in Figure 4 showing weight variation among the samples as a function of temperature. SWCNT paper was quite stable, while epoxy began to decompose around 400 °C under Ar gas conditions. The final residual material was 14.3 wt% for epoxy film (x) and 60.5 wt% for composite (y), based on which we determined that the SWCNT content of the composites was 46.2 wt% (y-x). As mentioned above, the space between SWCNT bundles in SWCNT paper contains a great deal of unfilled space. While the amount of epoxy impregnated in the SWCNT papers was insufficient to fill this space, we expect that it would be enough to effectively maintain the SWNCT network for the electrical conductivity of the SWCNT paper in the composites.

### 3.3. Electrical Conductivity and EMI SE

As the epoxy is an insulating polymer matrix, its use as the matrix of composites may adversely affect their electrical conductivity. A certain amount of epoxy is necessary if the shape of the SWCNT paper is to be maintained without disturbing the electrical conductivity of the SWCNT composites. In this study, we obtained 46 wt% SWCNT-reinforced epoxy composites using a three-step vacuum bagging process. The SWCNT paper and composite were both approximately 50 μm thick. Figure 5 reflects the similar electrical conductivities (2.78 × 10^4^ and 2.20 × 10^4^ S/m) of the SWCNT paper and composites. This result highlights how 46 wt% SWCNT-reinforced epoxy composites maintain the SWCNT paper’s electrical conductivity. Based on the 1-, 5-, and 10-layer prepregs, 46 wt% SWCNT-reinforced composites were fabricated at thicknesses of 50, 200, and 400 μm. The electrical conductivities of SWCNT paper and 1-, 5- and 10-layer composite materials were all maintained similarly while having an order of 10^4^ S/m. Therefore, SWCNT composite, which had electrical conductivity as high as SWCNT paper, was successfully fabricated by our tested manufacturing process.

The SWCNT composite’s high electrical conductivity was sufficient to achieve good EMI SE performance. Figure 6 shows the results of EMI SE and specific EMI SE at 1 GHz as a type of specimen. Although the SWCNT paper and one-layer composite were both 50 μm thick, the EMI SE result of the one-layer composite was lower due to the epoxy matrix, which acted as an insulator. As the thickness increased up to 400 μm, EMI SE was correspondingly enhanced, despite the similar electrical conductivity. It is a well-known fact that SE is enhanced as the thickness of a material is increased. To exclude this key fact, EMI SE was divided by density for results of specific EMI SE. Because the SWCNT paper did not contain epoxy, it showed relatively high specific EMI SE, more so than the composite, despite them being of the same thickness between SWCNT paper and one layer. Among composites, the 10-layer sample showed the highest value of the specific EMI SE, which makes sense, as this composite was the most compressed during the vacuum bagging system process. Zhang [33] et al. fabricated 1 wt% MWCNTs reinforced in epoxy composites to analyze the difference in specific EMI SE between the solid and foamed composites. The specific EMI SE was 5.2 dB cm^3^/g for the solid composite and 21.3 dB cm^3^/g for the foamed composite. In our study, the maximum specific EMI SE was 83.3 dB cm^3^/g in 46 wt% SWCNT-reinforced epoxy composite using the optimized vacuum bagging system. Based on these results, the high-loaded CNT-reinforced epoxy composite has an improved specific EMI SE. 

Studies have been conducted attempting to increase EMI shielding by using various fabrication processes, materials, morphologies and structures of composites, etc. [34,35,36]. Bagotia et al. [34] wrote a review paper based on studies that used MWCNTs as a filler when the polymer matrix was polycarbonate, and the EMI SE results were about 2.3~50 dB. Singh et al. [35] showed that the specific EMI SE of porous composites made of various shapes and materials is about 16~64 dB cm^3^ g^−1^. In the review paper of Wang et al. [36], the EMI SE of composites made of MWCNTs as a filler was 30~40 dB. Based on the results of various studies appearing in the review papers, the optimized vacuum bagging system in this study provides good quality of composites possessing a high EMI SE while being lightweight and thin.

We then determined valid thickness using the skin depth equation when SWCNT composites are applied in real-life. When electromagnetic (EM) waves emit towards the target material, some waves are reflected on the surface (reflections), while others are absorbed (absorptions). How the waves act depends on the target material’s surface. Skin depth is the penetration depth at which EM is absorbed and is expressed as follows:δ = 1/√(π × f × μ × σ)(1)
where δ is the skin depth, f is frequency (Hz), μ is permeability (H/m), and σ is electrical conductivity (S/m) [4,5,6,7]. As the frequency increases, permeability and conductivity are enhanced, and skin depth decreases exponentially. The calculated skin depth for frequencies ranging from 1.5 × 10^−2^ to 3.0 GHz is shown in Figure 7. As predicted, skin depth across all samples decreased exponentially as frequency increased. Skin depths at 1.0, 2.0, and 3.0 GHz are listed in Table 1. In SWCNT paper, the 1-layer composite was thinner than the skin depth, and the 5- and 10-layer composites were designed to be thicker than the skin depth. Where the composite is thinner than the skin depth, some EM waves are reflected and absorbed, but other remaining EM waves penetrate the material. In contrast, where the material is thicker than the skin depth, most EM waves are reflected at the material’s surface and absorbed by the surface of the conductive material [7]. Based on this fact, composites composed of five or more stacked layers act as efficient EMI SE materials. 

Finally, our SWCNT composite was folded into a complex shape (Figure 8a). Based on its high electrical conductivity, the SWCNT composite can be used to light LED lamps. The high-loaded SWCNT-reinforced epoxy composites produced by our efficient vacuum bagging process possess high electrical conductivity and EMI SE properties and potentially may have utility in industrial applications.

## 4. Conclusions

High-loaded 46 wt% SWCNT reinforced epoxy composites were fabricated by an optimized vacuum-bagging system under the temperature cycle depending on the time. This manufacturing process resulted in SWCNT composites with electrical conductivity equal to SWCNT paper. The SWCNT composite materials that shield most effectively against EMI are those that have high electrical conductivity and are of optimal thickness. The 10-layer composite produced the highest values of EMI SE (71.67 dB cm^3^/g) and specific EMI SE (83.3 dB cm^3^/g), both of which are a significant improvement over the results from the 1 wt% epoxy-reinforced MWCNT composites. These results confirm that the suggested vacuum bagging system is capable of producing good quality, lightweight, and optimal-thickness composites with a high EMI SE. 

We also calculated valid thickness based on the skin depth theory for thickness optimization. Based on our calculations, five-layer composites are the ideal thickness for EMI SE composites used for many applications. Our produced flexible and versatile composites were folded into complex shapes and, finally, used to light an LED lamp. 

## Figures and Tables

**Figure 1 polymers-13-01867-f001:**
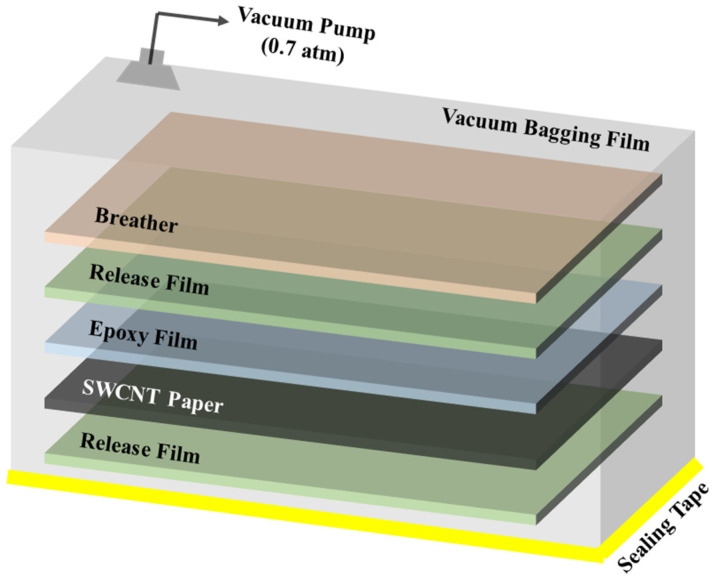
Illustration of our vacuum bagging system for the infiltration of semi-cured epoxy into SWCNT paper and complete curing.

**Figure 2 polymers-13-01867-f002:**
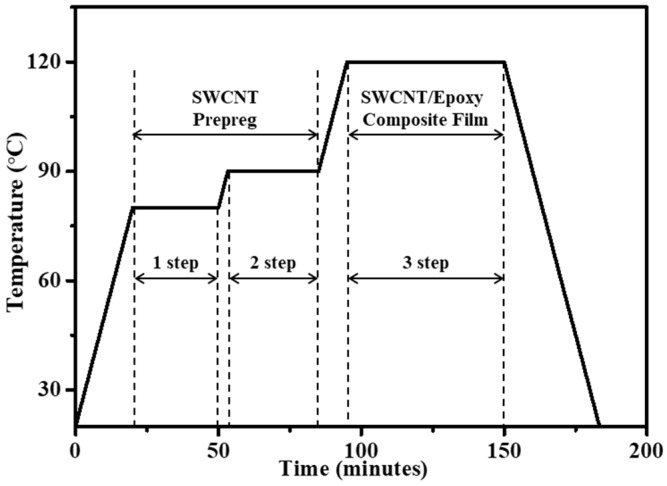
A 3-step cycle graph designed with temperature depending on the time.

**Figure 3 polymers-13-01867-f003:**
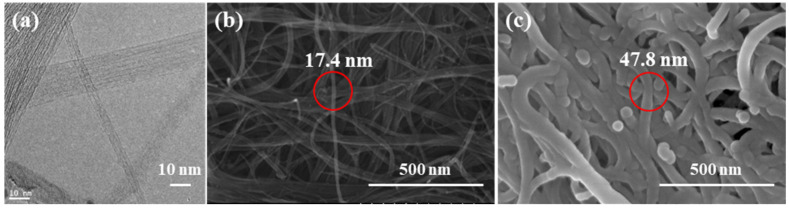
TEM and SEM images of (**a**) SWCNTs, (**b**) a surface of SWCNTs paper and (**c**) a surface of SWCNT/epoxy composite.

**Figure 4 polymers-13-01867-f004:**
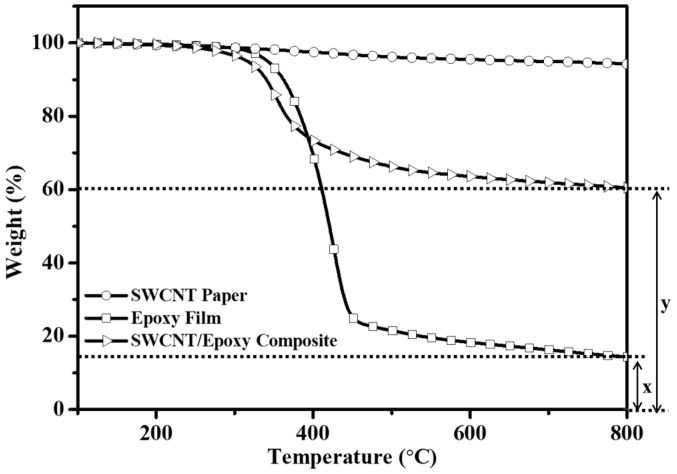
TGA plots of epoxy film, SWCNT paper, and SWCNT/epoxy composite; ‘x’ and ‘y’ are what remains after epoxy film and SWCNT/epoxy composite were burned, respectively.

**Figure 5 polymers-13-01867-f005:**
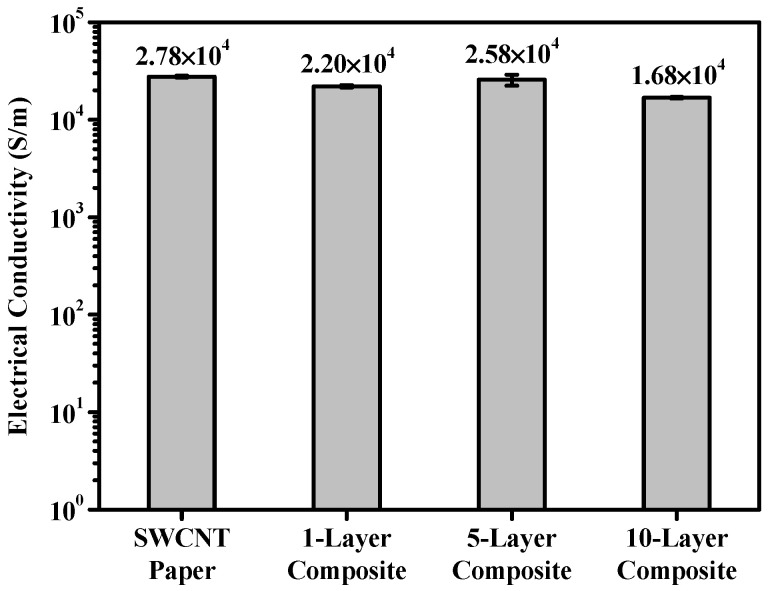
Electrical conductivity of SWCNT paper and stacked-layer composites.

**Figure 6 polymers-13-01867-f006:**
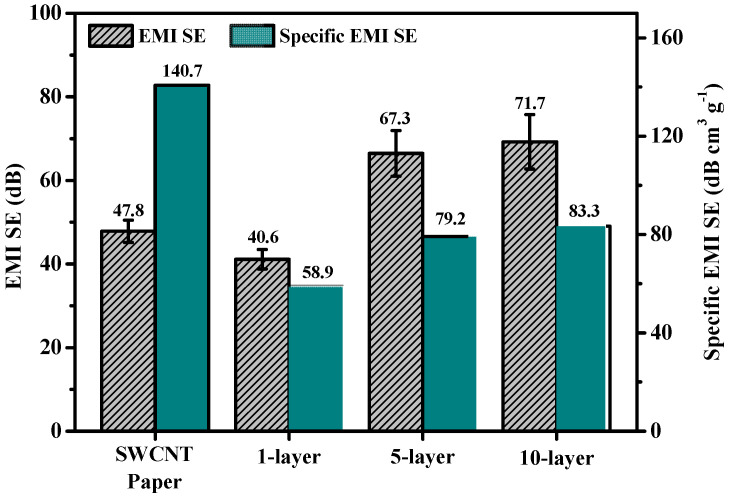
EMI SE and specific EMI SE results of SWCNT paper and 1-, 5-, and 10-layer composites.

**Figure 7 polymers-13-01867-f007:**
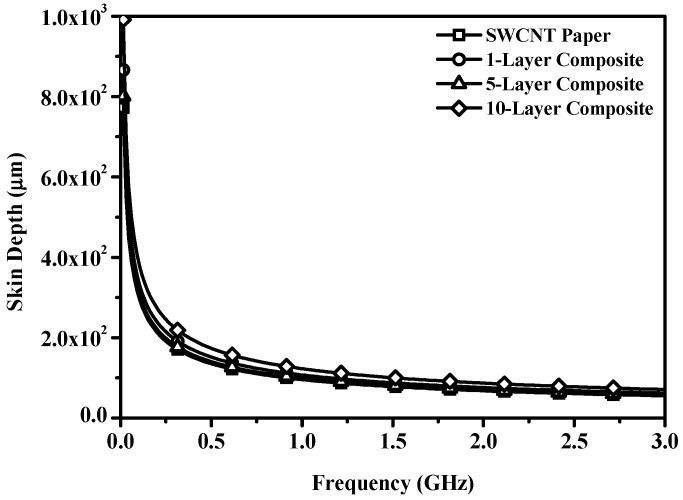
Calculated skin depth according to frequency of the SWCNT paper and 1-, 5-, and 10-layer composites.

**Figure 8 polymers-13-01867-f008:**
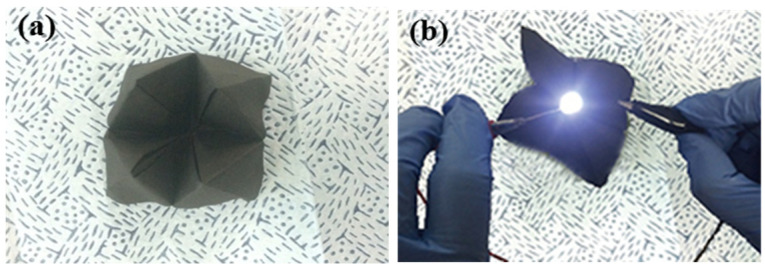
Photograph images of (**a**) flexible and folded SWCNT composite and (**b**) an LED lamp light shining through the SWCNT composite.

**Table 1 polymers-13-01867-t001:** Calculated values of skin depth (μm).

Frequency (Hz)	SWCNT Paper	1-Layer Composite	5-Layer Composite	10-Layer Composite
1.0	95.3	106.9	98.9	122.4
2.0	67.6	75.9	70.2	86.8
3.0	55.2	61.9	57.2	70.8
Thickness	50.0	50.0	200.0	400.0

## Data Availability

The data presented in this study are available on request from the corresponding author.

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
