# Peer review of "A Suggested Vacuum Bagging Process for the Fabrication of Single-Walled Carbon Nanotube/Epoxy Composites That Maximize Electromagnetic Interference Shielding Effectiveness"

_polymers, 2021, doi:10.3390/polym13111867_

Round 1
Reviewer 1 Report
The authors presented a work about manufacturing process for high loading SWCNTs reinforced in epoxy composites using a vacuum bagging system. The resulting products demonstrated high electrical conductivity.
Although the work is interesting in the field of nanomaterials, the impact, scientific significance and novelty of the study needs to be further elucidated in the introduction section; in addition, the authors have to justify why the manuscript is in the scope of polymers journal. Below are some detailed issues to be addressed before I can recommend publication::
- In the introduction section, the authors claim “Unfortunately, each of these processes suffer from some common disadvantages, namely that the weight concentration of resulting SWCNTs is not uniform, lacks reproducibility, and the processes aggregate the epoxy matrix between fillers and necessitates reliance on unclean production processes”, Therefore what is the goal in this research. because there is not evidence of reproducibility and repeatability data
- The authors should expand the introduction section as in its current state it does not provide enough background to demonstrate the impact and originality of their work.
- In the results section the authors must be described how many times and samples were measured in order to obtain the conductivity of the samples.
- Why only two pictures for SEM?, is necessary to provide a comparison between the samples and provide the size distribution of the SWCNT.
- A comparison with the literature table must be prepared with their results and performance of the layer composites.
Author Response
Thank you very much for your constructive comments. The manuscript has been revised as attached. We sincerely answered the reviewer's questions, and modified the manuscript following the reviewer's comments.

Reviewer 2 Report
Journal: Polymers
Title: A Suggested Vacuum Bagging Process for the Fabrication of High‐Loaded Single‐Walled Carbon Nanotube‐Reinforced Epoxy Composites that Maximize Electrical Properties and Electromagnetic Interference Shielding Effectiveness
Authors: Min Ye Koo et al.
MS No.: polymers-1228527
To whom it may concern,
The manuscript, “A Suggested Vacuum Bagging Process for the Fabrication of High‐Loaded Single‐Walled Carbon Nanotube‐Reinforced Epoxy Composites that Maximize Electrical Properties and Electromagnetic Interference Shielding Effectiveness” submitted by Min Ye Koo et al. is interesting. This work focuses on designing and manufacturing of composite material with maximized electrical conductivity and optimized electromagnetic interference (EMI) shielding effectiveness (SE). The manuscript can be accepted for publication after major revision. Some comments and suggestions are given below for improvement.
- If more single-walled carbon nanotubes was used, is it possible to further improve the performance of composite materials?
- The authors only introduced the purity of the single-walled carbon nanotubes used. Can the authors tell us about its coiled structure? Can the authors explain whether the SWCNT is a semiconductor type or a metal type? Do single-walled carbon nanotubes of different structures have the same performance of the composite materials?
- In section 2.2. How did the authors ensure the vacuum degree of the vacuum bagging process used?
- In Figures 4 and 6, please explain why the name Bucky Paper is used in the figure and the name SWCNT paper is used in the legend.
- The title of the manuscript is too long.
In general, the topic is interesting, but conclusions of the manuscript needs improving. Moreover, the English of the manuscript is not satisfaction and need mending. It is highly recommended a native English speaker modify the manuscript. Although the Polymers is an open access journal with high processing fee, it is still a peer-reviewed journal of polymer science published semimonthly online by MDPI. The manuscript in the current stage is not suitable for the publication in the esteemed journal and might be accepted after major revision.
Ruoyu Hong, Ph.D.
Minjiang Professor
Chem. Eng. Dept.
Fuzhou Univ.
Author Response

(The authors gave the same response as above.)

Reviewer 3 Report
The results presented by Koo et al in the manuscript could be useful for the improvement of this type of CNT-epoxy composites, could provide new information to the researchers in the field and have potential for practical applications. For these reasons, in principle, it is suitable for its publication in the journal Polymers. However, I kindly request for some amendments listed below before considering its acceptance:
- The title of the article is really long. I would gently recommend to shorten it.
- In the Introduction section, when the authors wrote: “EMI SE can also be enhanced by precisely controlling a material’s thickness, with thicker materials more effectively blocking EMI. In sum, the materials that shield most effectively against EMI are those that have high electrical conductivity and are thin thickness.” It looks like they claim contradictory things: “thicker materials more effectively blocking EMI” and later, they write that EMI shield is more effective if the material are “thin thickness”. Please, clarify this or rewrite the sentence.
- In the Introduction section, when the authors wrote: “To overcome these problems, various alternatives have been proposed. CNTs are generally dispersed in epoxy matrix through either chemical or physical methods [13]”. Please, add more related references, not only one and if they were recent, from the last 5 years, also it would be appreciated. These references should reflect the different chemical and physical methods the authors claim.
- In the section 2.2. Fabrications. Please: fabrication or fabrication methods.
- In the section 2.2. Fabrications, quoting the authors: “The material made during steps 1 and 2 (referred to as ‘SWCNT’ prepreg in this study) Epoxy in this step had a low viscosity as a result of the temperature and pressure it was exposed to,”. Please, correct and rewrite the sentence.
- In the section 2.2. Fabrications, please, explain better the three steps of the proposed vacuum bagging methods. The way it is currently written is confusing or not clear enough.
- In section 3.1. Morphology, quoting the authors: “However, randomly‐entangled SWCNT bundles can cause desirable air fractions, and impair the electrical properties of the SWCNT paper or composite [28, 29]”. The air fractions are NOT desirable, right? If so, please, correct this.
- In the section 3.3. Electrical conductivity and EMI SE, with respect to figure 5, the authors wrote: “While electrical conductivity decreased slightly as the number of stacked prepregs went higher,”. In figure 5, I see that the electrical conductivity increases with 5 layers and decreases with 10 layers. What is the reason behind? Could the authors elaborate with more detail this observed result, beyond what they have already explained in the current manuscript?
Author Response

(The authors gave the same response as above.)

Round 2
Reviewer 1 Report
The authors attended all the observations.
Author Response
Thanks for your kind advice. In the 2nd revised manuscript submitted this time, some words have been revised to express the content more clearly. Thanks again for the advice.
Reviewer 2 Report
The authors already revised the manuscript extensively. The manuscript could be accepted for publication after minor revision. The English of the manuscript could be improved.Author Response
Thanks for your kind advice. In the 2nd revised manuscript submitted this time, some words have been revised to express the content more clearly. Thanks again for the advice.
Reviewer 3 Report
I really appreciate the effort made by the authors in order to amend all the points I have asked for. I have read both their reply and the latest manuscript carefully again and I think it has improved substantially to be accepted for publication in the journal Polymers. Now the article has more information about the processing and the used materials and the fabrication steps are explained in more detail. They also have amended some figures for the sake of clarity to the readers.
Author Response

(The authors gave the same response as above.)
